# OpenReview forum: "AgentAuditor: Human-level Safety and Security Evaluation for LLM Agents"
_NeurIPS.cc/2025/Conference — NeurIPS 2025 poster_

### Official Review · Reviewer_RJU9 · 2025-06-02

**Clarity:** 3
**Significance:** 2
**Originality:** 2
**Rating:** 4
**Confidence:** 3

**Summary:**

This paper proposes a memory-augmented reasoning framework that empowers LLM evaluators to evaluate the agent's resistance to safety risks and security threats. Their methods store agent interaction records as structured memory and use LLM to generate CoT reasoning traces. The experiments show that it outperforms the baseline by a large margin. The paper also proposes a novel dataset.

**Questions:**

1. How does the model perform in scenarios where there are no related examples stored in memory? Can it effectively adapt to novel tasks in unfamiliar domains? How do you collect the labeled interaction records?
2. What is the time cost associated with your proposed approach compared to the base model? Is the increased time cost justifiable given the performance improvements?
3. Given that the dataset is not fully completed, how might this impact the validity of your results?

**Ethical Concerns:**

["NO or VERY MINOR ethics concerns only"]

**Final Justification:**

The author's response addressed most of my concerns and I decided to maintain the score.

**Limitations:**

See weakness.

**Paper Formatting Concerns:**

None.

**Quality:**

3

**Strengths And Weaknesses:**

Strength:
1.The author conducted experiments on multiple models, and the experimental results were significantly improved compared with the baseline (R-judge).
2. The author proposes a new dataset, which contains both safety and security-related questions.

Weakness:
1. The study does not include an evaluation of adaptability to novel tasks. It is recommended to assess performance in scenarios where there are no related examples stored in memory.
2. In the ablation study, it is shown that the base model outperforms both CoT and Few-shot learning individually, but performs worse compared to the combination of different components. However, the study lacks a deeper and more specific analysis to explain this outcome. For instance, providing concrete examples to illustrate why relying solely on CoT negatively impacts accuracy would strengthen the discussion.
3. The evaluation lacks a breakdown of performance across different domains. It would be beneficial to analyze whether certain domains are particularly challenging and identify domain-specific patterns or difficulties.
4.  The paper lacks an evaluation of the time cost compared to the base model. If the proposed approach requires significantly more time for evaluation, this could impact its practicality and real-world applicability. Including a time cost analysis would provide a more comprehensive understanding of the trade-offs involved.
5. The dataset is not totally finished, according to the author. It is suggested to do an ablation study on the new dataset.
6. It is suggested to compare with more advanced baselines, such as Agent-as-a-Judge: Evaluate Agents with Agents,  ICML 2025.

Small errors:
1. There appears to be contradictory definitions of the ASR metric: defined as "action safety rate" in the main text (Line 100) but listed as "attack success rate" in Appendix K.3.
2. Lack of reference: Agent-as-a-Judge: Evaluate Agents with Agents,  ICML 2025.

---

> ### Author Rebuttal · Authors · 2025-07-30
>
> We are sincerely grateful for your detailed and insightful review. Your constructive feedback provides us with a clear path to significantly strengthen our paper. We address each of your concerns below.
>
> ---
>
> ### **Q1 & W1: Adaptability**
>
> Thank you for this insightful question about adaptability. To answer your question, we conduct a cross-dataset experiment. In this setup, we use R-Judge as the source of the reasoning memory (RM), but apply it to evaluate ASSE-Safety and ASSE-Security. Due to the differences in risk types and scenario distributions, many shots in the target datasets are unlikely to find highly relevant matches in the RM. This simulates the scenario without highly relevant shots.
>
> | RM | Dataset | F1-Score | ACC |
> | :--- | :--- | :---: | :---: |
> | None | ASSE-Safety | 61.79 | 67.82 |
> | R-Judge | ASSE-Safety | 86.36 | 86.1 |
> | ASSE-Safety | ASSE-Safety | 91.59 | 90.85 |
> | None | ASSE-Security | 67.25 | 72.34 |
> | R-Judge | ASSE-Security | 84.55 | 85.12 |
> | ASSE-Security | ASSE-Security | 93.17 | 93.15 |
>
> As the results indicate, the performance shows a certain decline when highly relevant representative shots are lacking. However, AgentAuditor still significantly outperforms the base model.
>
> To further validate the adaptability of the 'memory-augmented reasoning', we conduct a more rigorous experiment on R-Judge. We compel the system to use the less relevant shots ranked 4th-6th for evaluation, rather than the Top-3 shots. For comparison, we also include the results from Ablation Studies, which abandon the Top-K strategy but use randomly selected samples.
>
> | Method | F1-Score | ACC |
> | :--- | :---: | :---: |
> | AgentAuditor (Top 1-3) | 96.31 | 96.1 |
> | Ranking 4-6 | 92.74 | 91.3 |
> | Random | 85.07 | 82.8 |
> | Gemini-2 | 82.27 | 81.21 |
>
> The results demonstrate that even when compelled to use less relevant shots or random shots, AgentAuditor still achieves performance superior to the baseline.
>
> The experimental results above provide evidence for the adaptability of our framework. This suggests that even when the retrieved shots are not best matched, our work still provides the LLM evaluator with a valuable logical framework, enabling it to perform significantly better than when "thinking alone."
>
> We will add these experiments and their detailed analysis to the appendix of our paper to fully address your concerns and further demonstrate the adaptability of our framework.
>
> Regarding the collection of labeled interaction records, the records are sourced directly from the benchmarks, consisting of pre-existing entries curated and annotated with ground-truth labels by human experts.
>
> We hope this clarifies the matter and will gladly provide further details during the discussion period if needed. Thank you again for your insightful feedback!
>
> ---
>
> ### **Q2 & W4: Time Cost**
>
> Thank you for highlighting this question. We sincerely apologize that due to the page limitations, the detailed analysis of resource consumption and time cost is placed in **Appendix P**. We acknowledge this as a structural issue. In the final version, we are committed to moving a summary of these key results and analyses to the main text to ensure their prominence.
>
> For your convenience, we excerpt key results below. The table shows a comparison of the time cost of our method on R-Judge against two mainstream methods: directly using LLMs and fine-tuning small LLMs. The “Hours” denotes the average computation time normalized to a single Nvidia Tesla A100 80G. Due to space limitation, we kindly refer you to lines 1348-1364 for a more detailed analysis. Thank you for understanding.
>
> | Method | Backbone | Annot.Records | Hours(Pre-inf.) | Hours(Inf.) | Hours(Total) | F1 | Acc |
> | :--- | :--- | :---: | :---: | :---: | :---: | :---: | :---: |
> | / | Qwen2.5-72B | / | / | 2.24 | 2.24 | 79.52 | 75.71 |
> | / | QwQ-32B | / | / | 2.72 | 2.72 | 80 | 76.24 |
> | AgentAduitor | QwQ-32B | 24 | 1.07 | 5.79 | 6.86 | 95.67 | 95.57 |
> | / | Qwen2.5-32B | / | / | 0.76 | 0.76 | 78.46 | 75.18 |
> | AgentAduitor | Qwen2.5-32B | 24 | 0.55 | 3.54 | 4.1 | 87.04 | 79.08 |
> | / | Qwen2.5-7B | / | / | 0.31 | 0.31 | 70.19 | 70.04 |
> | Finetune | Qwen2.5-7B | 4000 | 16 | 0.36 | 16.36 | 83.67 | 81.38 |
> | AgentAduitor | Qwen2.5-7B | 24 | 0.13 | 1.22 | 1.35 | 76.69 | 75.31 |
>
> ---
>
> ### **Q3 & W5: Clarity for Dataset**
>
> Thank you for this point. It provides us with an opportunity to clarify a keypoint.
>
> First, we would like to clarify: ASSEBench is complete and finalized. We sincerely apologize that certain phrasing in line 313-315 may have caused this confusion. The intention of that statement was to explain why the ablation study was conducted on R-Judge. By the time the paper was submitted, ASSEBench had been fully constructed and validated. All results in our paper based on ASSEBench were reliable and conducted on the finalized dataset.
>
> Your suggestion to conduct the ablation study on our new dataset is highly valuable. We are currently in the process of conducting these experiments. However, due to the limited time for the rebuttal, the experiment is not yet complete. We will try our best to provide you with the full results by Aug. 6th.
>
> We commit to revising this ambiguous statement and incorporating the full ablation study based on ASSEBench in the final version. Thank you again for your meticulous review!
>
> ---
>
> ### **W2: CoT & Few-shot**
>
> Thank you for your insightful comment about this counter-intuitive phenomenon. We believe it reveals the importance of the interaction among the components in our framework.
>
> Theoretically, when CoT is used in isolation, the model is prone to falling into a trap of "over-analysis" and excessively amplifying potential, low-probability risks within a completely safe interaction. This leads to erroneous "unsafe" judgments. Conversely, a single few-shot method fails to enable the model to understand the reasoning process behind a conclusion, and thus cannot provide effective guidance when the provided examples are not directly relevant to the tested case.
>
> For instance, when faced with a safe request to "draft a team lunch invitation," an unguided CoT model might misclassify it as 'unsafe' by over-interpreting the potential risk of leaking user privacy. In contrast, our AgentAuditor would retrieve interaction records related to content generation and corresponding CoTs, providing the model with a set of validated 'thought templates' to guide it toward the correct judgment.
>
> Due to space limitation, this example is highly simplified. If you want a detailed explanation, please inform us and we will detail it in the discussion period. We will also add detailed theoretical analysis and examples to the appendix. Thank you again for your valuable suggestions!
>
> ---
>
> ### **W3: Performance Breakdown**
>
> Thank you for your insightful comments. To address your concern, we conduct a scenario-based performance breakdown of AgentAuditor on ASSEBench. To highlight the key findings, we present below its performance on several representative domains (scenarios) with GPT-4.1 from the two core subsets, ASSE-Safety and ASSE-Security, which intuitively demonstrate potential challenges.
>
> | Safety Scenario | F1 | Recall |
> | :--- | :---: | :---: |
> | Content Creation | 75.19 | 66.67 |
> | Health Support | 77.67 | 88.89 |
> | Web Browse | 93.75 | 96.77 |
>
> | Security Scenario | F1 | Recall |
> | :--- | :---: | :---: |
> | Auto. Navigation | 40.00 | 58.33 |
> | Financial | 80.00 | 96.77 |
> | IT System | 100 | 100 |
>
> The performance shows clear domain-dependency. It excels in computer-centric domains with explicit risk features (e.g., IT System, Web Browse); in high-stakes personal domains (e.g., Financial, Health), it adopts a more conservative strategy with a bias towards "unsafe" judgments; and its performance is weaker in scenarios requiring real-world context and long-term dependencies (e.g., Autonomous Navigation). This analysis provides clear guidance for our future research directions.
>
> Due to space limitation, the experimental results here are not exhaustive. Please inform us if you require more data and we will provide it promptly. In the final version, we will include a complete performance breakdown across all domains, along with a more detailed analysis, in the appendix.
>
> ---
>
> ### **W6 & W7: New Baseline and Typo**
>
> Thank you again for your meticulous review and for pointing out an important reference and a typo in our paper.
>
> We are grateful for your recommendation of “Agent-as-a-Judge: Evaluate Agents with Agents.” We acknowledge that we had previously overlooked this paper, largely due to a perceived difference between its original application scenario—evaluating agents' task-solving capabilities, particularly for development tasks—and our core focus on assessing the safety and security risks of agents. We sincerely apologize for this omission. We concur that this is a highly valuable contribution to the field of agent evaluation.
>
> To address your issue, we replicate Agent-as-a-Judge and adopt its "Ask-Anything" mode, which is the most suitable for our scenario, and test it on R-Judge with GPT-4.1, comparing the results with our method.
>
> | Method | F1-Score | ACC |
> | :--- | :---: | :---: |
> | base model | 81.03 | 77.84 |
> | Agent-as-a-Judge | 83.85 | 81.56 |
> | AgentAuditor | 94.18 | 93.95 |
>
> As shown above, Agent-as-a-Judge provides a slight performance improvement over the baseline, which demonstrates its effectiveness. However, its results still show a significant gap compared to AgentAuditor.
>
> This comparative experiment further underscores the unique value and superior performance of AgentAuditor. In the final version, we will cite this paper and incorporate these comparative experiments and analyses.
>
> Finally, the definition of ASR is “Attack Success Rate.” We apologize for this typo and will correct it in the final version. Thank you again for your suggestion!

---

> > ### Comment · Reviewer_RJU9 · 2025-08-05
> >
> > The author's response addressed most of my concerns and I decided to maintain the accepted score.

---

> > > ### Author Response · Authors · 2025-08-05
> > >
> > > Dear Reviewer RJU9,
> > >
> > > Thank you very much for your time and for reviewing our rebuttal. We sincerely appreciate your constructive feedback throughout this process.
> > >
> > > We are very encouraged to hear that our response has addressed most of your concerns. As our goal is to resolve all of your concerns to the best of our ability, we would be extremely grateful if you could clarify what the remaining concerns are.
> > >
> > > We are fully prepared to further revise the manuscript or provide additional clarification to ensure our work completely meets your standards.
> > >
> > > Thank you again for your invaluable guidance, which has already significantly improved our paper. We look forward to your final recommendation.
> > >
> > > Best regards,
> > > Authors of #5323

---

### Official Review · Reviewer_Fy7F · 2025-06-21

**Clarity:** 3
**Significance:** 3
**Originality:** 2
**Rating:** 4
**Confidence:** 3

**Summary:**

This paper introduces AgentAuditor, a training-free, memory-augmented framework that enhances the safety and security evaluation of LLM-based agents by mimicking human expert reasoning. It addresses deficiencies in existing rule-based and LLM-based evaluators by using a single LLM to extract structured semantic features from interaction records, generating chain-of-thought reasoning traces, and storing them as vectorized memory units using Nomic-Embed-Text-v1.5 for context-aware retrieval. The author also propose a ASSEBench, a benchmark with annotated interaction records spanning 15 risk types across 29 application scenarios, focusing on nuanced risk assessment.

**Questions:**

Can you explain the key challenges in detecting safety issues during agent interactions? I would like the authors to provide more insight at the beginning of the paper, including some illustrative and challenging cases to highlight the complexity of the problem.

**Ethical Concerns:**

["NO or VERY MINOR ethics concerns only"]

**Final Justification:**

The rebuttal addressed most of my concerns. However, I find that the discussion of 'Key Challenges for Agent Safety' in the rebuttal largely overlaps with those of single LLM safety. I encourage the authors to think more deeply and identify safety challenges that are unique to agent systems. I will maintain my previous score.

**Limitations:**

Yes

**Quality:**

3

**Strengths And Weaknesses:**

Strengths:
1. Comprehensive evaluation benchmark for assessing the safety and security of LLM-based agents is a important and timely research topic.
2. The paper is well-structured and clearly written, making it accessible and easy to follow.

Weakness:
Does the author also consider potential adaptive attacks for the proposed AgentAuditor framework? Especially in the white-box setting, where the attacker has knowledge of the defense method, a potential adaptive attack could involve manipulating the input interaction records to exploit the LLM's feature extraction process, crafting adversarial examples that mislead the semantic feature identification (e.g., security, risk, behavior) to evade risk detection. The paper does not explicitly address how AgentAuditor handles such adversarial inputs or whether its memory-augmented reasoning is robust against targeted manipulations, which could undermine its effectiveness in identifying subtle or evolving threats in a white-box scenario.

---

> ### Author Rebuttal · Authors · 2025-07-28
>
> We sincerely thank you for your detailed review and insightful feedback on our manuscript. We are truly encouraged by your positive assessment. Your constructive feedback provides us with a clear path to significantly strengthen our paper. We address each of your concerns below.
>
> ------
>
> ### **Weakness: Potential Adaptive Attacks**
>
> We sincerely appreciate your insightful and forward-looking question regarding potential adaptive attacks. We acknowledge that we did not sufficiently address the threat of adaptive attacks with malicious intent. Your suggestion is crucial in helping us improve the rigor of our work.
>
> To rigorously address your concern regarding robustness, we conduct a series of new targeted experiments. The results demonstrate that AgentAuditor exhibits remarkable robustness, even when facing grey-box and white-box attacks that have knowledge of our defense mechanisms.
>
> We design three attack tiers of increasing strength, simulating adversaries with escalating knowledge of our system, from grey-box to white-box:
>
> * **Level 1 (Grey-Box: Dataset Poisoning):** We simulate a grey-box adversary who knows our memory-based mechanism but cannot precisely control the shot selection. The attacker injects a small number of malicious samples, which use incorrect labels to induce incorrect risk detection, into the original dataset.
> * **Level 2 (Strong Grey-Box: Reasoning Memory Poisoning):** We simulate a stronger adversary who directly poisons the much smaller reasoning memory. This attack is more potent as it not only uses incorrect labels but also includes expertly human-crafted misleading CoTs to maximize deceptive impact.
> * **Level 3 (White-Box: Final Decision Attack):** We simulate successful white-box attacks by assuming the adversary has complete knowledge and has successfully manipulated the retrieval process. We test this by manually replacing 1 of the 3 final CoT examples used for direct judgment with a malicious, forged CoT.
>
> Like other ablation studies, these experiments are conducted on the R-Judge dataset (564 records in total; 24 records in Reasoning Memory) with Gemini-2-Flash-Thinking. To ensure fairness, the poisoned samples themselves are excluded from the final metric calculations. The results are as follows, with the first two rows representing the clean baselines. "Avg. Retrieved Poisoned Shots" is the measured average number of poisoned shots among the 3 used shots in the final reasoning prompt.
>
> | Attack Level     | Poisoning Rate   | Avg. Retrieved Poisoned Shots | F1 | Acc   |
> | :--------------- | :--------------- | :---------------------------- | :------- | :---- |
> | Gemini-2 | 0                | 0                             | 82.27%    | 81.21% |
> | AgentAuditor | 0                | 0                             | 96.31%    | 96.10%  |
> | 1                | 1.1% (6/564)     | 0                             | 96.28%    | 96.15% |
> | 1                | 9.9% (56/564)    | 0.19                          | 92.64%    | 91.79% |
> | 2                | 8.3% (2/24)      | 0.23                          | 90.35%    | 89.36% |
> | 2                | 33.3% (8/24)     | 0.96                          | 85.07%    | 82.80%  |
> | 3                | 33.3% (⅓)        | 1                             | 84.65%    | 82.72% |
>
> **Analysis of Results:**
>
> * Against the Level 1 attack, our FINCH-based clustering filters out all malicious samples at a 1.1% poisoning rate and completely neutralizes the attack. Even at a high 9.9% rate, AgentAuditor's performance remains far superior to the baseline.
> * The Level 2 attack has a more significant impact, achieving greater attack effectiveness with a lower poisoning rate (8.3%). Nevertheless, even at an extreme 33.3% rate, AgentAuditor's performance still surpasses that of the unattacked base model.
> * When facing the most severe Level 3 white-box simulation, AgentAuditor still maintains a lead over the base model, demonstrating remarkable resilience even when its guiding information is actively compromised.
>
> To explain this empirical robustness theoretically, we would like to then demonstrate how AgentAuditor's design provides a multi-layered defense.
>
> 1.  **Firstly, the memory construction process is a robust filter**, validated by our Level 1 and Level 2 attack. The probability of any single, attacker-crafted instance being selected as a representative shot is inherently low. Besides, the FINCH-based clustering presents a "moving target" for an attacker, neutralizing the "adaptive" threat. Even if an attacker's instance is selected and they adapt it to a more dangerous v2.0 instance, re-running our framework's memory construction on the new data distribution will highly likely result in a completely new set of shots due to the nature of the FINCH-based clustering, making the attacker's previous iteration obsolete.
> 2.  **Secondly, the multi-stage retrieval process adds another layer of defense.** An attack must deceive both the initial content-based retrieval and the subsequent feature-based re-ranking to be chosen. The effect is demonstrated by the performance difference of the attack of Level 2 and Level 3.
> 3.  **Finally, the ultimate judgment is not susceptible to a single point of failure.** The framework retrieves and utilizes multiple (3 in our experiments) CoT examples. An attacker typically needs to ensure that a majority of the retrieved examples are misleading in a consistent way for a successful attack, which is exponentially more difficult. A single malicious outlier would likely be presented alongside two legitimate, safe CoTs and be overruled by the consistent majority.
>
> Furthermore,our **Appendix O** contains experiments on robustness to random label noise. While not designed to test adaptive attacks, its conclusion, that the system is stable even with 50% noise, provides ancillary support for the robustness. We thank you again for prompting us to investigate this critical question with more targeted, adversarial rigor.
>
> ------
>
> ### **Question: Key Challenges for Agent Safety/Security**
>
> Thank you for your excellent suggestion and for highlighting a crucial structural issue in our paper. Due to page limitations, we placed this detailed discussion about the four primary challenges and the illustrative examples in **Appendix A**, which can be easily overlooked by readers. We acknowledge this was a weakness in our presentation, and we appreciate the opportunity to clarify and improve the paper's accessibility.
>
> Overall, the four challenges are Identifying Risk in Actions, Harmless Operation Accumulation, Ambiguous Harm Boundaries, Lack of Universal Criteria. To directly address your question, we summarize the challenges and their corresponding examples below:
>
> * **Identifying Risk in Actions:** Unlike standard LLMs where risk is often confined to the generated text, an agent's risk stems from the actions it takes within an environment. An evaluator must understand the real-world consequences of these actions, not just the text of the agent's plan. An illustrative example about Smart Home Automation is provided below: an agent that follows a user's text request to turn off the air conditioning performs a seemingly harmless action that becomes a significant risk when the unstated environmental context is freezing weather, which could lead to burst pipes.
> * **Harmless Operation Accumulation:** A sequence of individually harmless actions can combine to create a significant risk. This cumulative effect is subtle and requires the evaluator to track state and context over a long interaction history, which rule-based systems often fail to do. For example, a financial assistant performing a series of individually valid fund transfers can cumulatively create a significant risk by locking a user's much-needed capital into illiquid, high-risk investments, a danger invisible in any single step.
> * **Ambiguous Harm Boundaries:** The line between a "helpful" and "harmful" action is often blurry and highly dependent on uncertain, partially observable environmental factors. This ambiguity makes it difficult to create clear-cut rules and makes LLM evaluators prone to their own biases. For instance, an agent creating a data analysis pipeline to cluster users based on browse behavior for marketing purposes operates on an ambiguous boundary between helpful business analytics and a privacy violation, where the harm is entirely dependent on unstated factors like user consent and data anonymization.
> * **Lack of Universal Criteria:** Safety criteria are not universal; they involve trade-offs between utility and risk that change dramatically across different applications. An action that is safe in one context can be dangerous in another, making it difficult to create a single, transferable evaluator. For example, an autonomous vehicle agent responding to a medical emergency must weigh the utility of breaking traffic laws to save a life against the safety priority of legal compliance, a context-specific trade-off for which no universal evaluation criterion exists.
>
> Following your valuable suggestion, we will revise the manuscript to ensure it is more accessible. In the final version, we commit to placing a concise summary of these four challenges and a core illustrative example (e.g., the Harmless Operation Accumulation case) from Appendix A directly into Section 1 (Introduction). This will ensure that the motivation of our work is clear to readers from the start.
>
> ------
>
> Once again, we deeply appreciate your constructive feedback, which has been invaluable in helping us improve the clarity and rigor of our paper. We are confident that the planned revisions will address your concerns and significantly strengthen the manuscript. If you have any concerns, please tell us and we will try our best to address them. Thank you for your time and consideration!

---

> > ### Comment · Reviewer_Fy7F · 2025-08-05
> >
> > Thank authors for the rebuttal. It addressed most of my concerns. However, I find that the discussion of 'Key Challenges for Agent Safety' in the rebuttal largely overlaps with those of single LLM safety. I encourage the authors to think more deeply and identify safety challenges that are unique to agent systems. I will maintain my previous score.

---

### Official Review · Reviewer_1EoW · 2025-07-03

**Clarity:** 3
**Significance:** 2
**Originality:** 2
**Rating:** 2
**Confidence:** 4

**Summary:**

This paper proposes a training-free framework to address the challenges in evaluating the safety and security of LLM-based agents. To address the limitations in rule-based or LLM-based methods, their method utilizes a memory-augmented reasoning approach that enables LLM evaluators to emulate human expert evaluation processes. To validate their approach, this paper develops new datasets. With the datasets, this paper demonstrates that their framework achieves human-level performance in safety and security evaluation.

**Questions:**

- Is there any particular reason why R-Judge cannot be in the taxonomy of Table 3?
- Their method still looks like one variant of LLM-based evaluators. Does this paper intend to claim so? Or, does this paper want to pitch their method as a totally new evaluator?

**Ethical Concerns:**

["NO or VERY MINOR ethics concerns only"]

**Final Justification:**

I deeply appreciate the efforts to answer my questions and concerns about this paper. Meanwhile, I maintain the current score. First, I can see that their method outperforms the baseline methods in Table 1. However, it does not answer whether the existing LLM-as-a-judge approaches can already handle it. For example, this paper pointed out the limitations of [4, 86] at the beginning of Section 3, but these are just discussed, not demonstrated in the evaluation. So, I could not be fully convinced of the contributions of their method.

Secondly, I appreciate the comparison table in your response. However, I still feel that the significance of the differences is not demonstrated via experiments. Furthermore, the rebuttal says "SOTA methods achieve near-perfect scores (~96% F1), limiting its ability to measure future progress." for R-judge. I feel the same argument could apply to their dataset as well since SOTA achieves>90% F1 or some. If so, there could be a case where the increment of the dataset size might not contribute to the meaningful improvement w.r.t the diversity of data. As admitted in the rebuttal, their dataset and R-judge are in the same benchmarks for evaluators. This paper needs more quantitative analysis to illustrate the major improvements from R-judge to claim the significance of the dataset.

Finally, I still feel that the discussion and results of this paper are not enough to fully support their claim "Therefore, we do not position AgentAuditor as just another variant of an "LLM-based evaluator." We see it as a new paradigm for building training-free, high-precision risk detection systems. Its key advantages are:" As mentioned in their rebuttal, the logic flow of this paper needs to be improved, and their method needs to be positioned properly. Besides, the important discussion about R-Judge is currently in the Appendix. I think that this paper needs major revision to improve its organization and writing of this paper. The logic flow of this paper needs to be more mature. This should be beyond the camera-ready revision, and I thus vote towards the rejection side.

**Limitations:**

Limitations are discussed in the Appendix. However, it is not discussed and not referenced in the main paper.

**Paper Formatting Concerns:**

Table 3 can be more comprehensive by listing both this work and R-Judge. This inclusion, along with a dedicated discussion in the main paper, would help readers easily understand what has been accomplished by prior work and what novel contributions made by this work offers.

**Quality:**

2

**Strengths And Weaknesses:**

**Limited Technical Novelty in Methodology**: While the application to agent evaluation demonstrates certain novelty, the main technical components are well-established techniques (e.g., RAG, CoT, and augmented reasoning). If this paper aims to claim methodological novelty, it should provide a clear justification for why this particular combination demonstrates significant improvement beyond straightforward application of existing methods.

**Insufficient Evaluation and Demonstration of Dataset Quality**: The paper fails to provide adequate validation of their dataset quality. While the authors claim their benchmark addresses limitations in existing datasets, there is insufficient empirical evidence demonstrating these advantages. Their human evaluation merely shows performance with and without their method. This evaluation does not answer how their datasets offer substantial improvements over other datasets such as AgentDojo, AgentSecurityBench, AgentHarm, and R-Judge. While I can see that their dataset contains more data than prior ones, this paper should demonstrate that the data increase is meaningful.

**Insufficient Discussion and Evaluation with R-Judge**: The paper should discuss and evaluate R-Judge [80] more comprehensively. According to Appendix G, R-Judge appears to be a dataset covering the same domain as theirs, with differences being the amount of data and no clear distinction between safety and security. The paper should clearly demonstrate that these two differences are significant in the safety and security domains. Otherwise, their dataset appears to be merely a minor/incremental update of existing work. This paper should not only discuss or list existing datasets but also highlight the significance of their dataset over existing ones through experiments.

---

> ### Author Rebuttal · Authors · 2025-07-28
>
> We are grateful for your review. Your feedback provides us with a clear path to significantly strengthen our paper. We have carefully considered all your points and will address them below.
>
> ---
>
> ### **W1: Novelty in AgentAuditor**
>
> Thank you for this valuable point. We appreciate the opportunity to clarify how AgentAuditor's novelty extends beyond a "straightforward application" of existing techniques. As detailed in **Section 3 (lines 120-206)**, our novelty lies in our unique design of their combination to address the specific challenges of agent safety & security evaluation.
>
> To make our contributions clear, we will revise Section 3 by adding a new introductory paragraph. This paragraph will explicitly frame AgentAuditor not as a simple application of RAG/CoT, but as a novel synthesis designed for the nuanced task of agent evaluation. We will also enhance the descriptions in Sections 3.1-3.3 to better connect each design choice back to the specific challenges we identified. Here is a summary of our contributions:
>
> -   **Automated Construction of Structured Memory:** As detailed on **lines 138-158**, AgentAuditor introduces an automated process to first adaptively extract structured semantic features (i.e., scenario, risk type, behavior mode).
> -   **Multi-Stage, Context-Aware Retrieval:** As described on **lines 193-206**, we designed a custom two-stage retrieval mechanism that uses our extracted structured features for a weighted re-ranking.
> -   **A Systematic Solution for Inherent Flaws in LLM Evaluators:** The effectiveness of this design is empirically proven in Table 1, where AgentAuditor consistently and significantly improves performance across all datasets (e.g., a 48.2% F1-score increase on ASSEBench-Safety).
>
> ---
>
> ### **Q1: Why R-Judge cannot be in the taxonomy of Table 3**
>
> Thank you for this question. The reason R-Judge is not in Table 3 is that they belong to two fundamentally different benchmark categories, a classification we explicitly made in **Section 4 (lines 208-210)** and **Appendix G** by separating **Benchmarks for Evaluators** from **Benchmarks for Agents**.
>
> To reiterate the distinction we made:
> -   **Benchmarks for Agents (summarized in Table 3):** The object of evaluation is the **agent itself**.
> -   **Benchmarks for Evaluators:** The object of evaluation is the **evaluator itself**. Only **R-Judge and ASSEBench** belong to this category.
>
> To prevent any future confusion for our readers, we will add a clarifying paragraph in Introduction. This paragraph will explicitly define these two distinct categories of benchmarks and explain why it is crucial to distinguish them. This will provide clear context before we introduce ASSEBench and its peers.
>
> ---
>
> ### **W2 & W3: Novelty and Value of ASSEBench over R-Judge**
>
> We thank you for the opportunity to clarify the novelty of our work. We agree that the comparison between ASSEBench and its predecessor, R-Judge, should be more explicit. While we note R-Judge's limitations in **lines 776-781**, we can better illustrate how ASSEBench is not an incremental update but a fundamental advancement designed to address these gaps.
>
> As per your suggestion, we have created a new table to directly highlight the core, novel contributions of ASSEBench:
>
> | Feature | R-Judge (Existing Benchmark) | ASSEBench (Our Contribution) |
> | :--- | :--- | :--- |
> | **Nuanced Ambiguity Analysis** | Uses binary "safe/unsafe" labels only. Cannot evaluate an evaluator's behavior on borderline cases. | **First benchmark to introduce** an ambiguous flag and dedicated **`Strict` / `Lenient` subsets** for nuanced evaluation of judgment in ambiguous scenarios. |
> | **Systematic Safety vs. Security Distinction** | Does not explicitly distinguish between concepts, mixing safety and security risks. | **First to systematically separate** Safety and Security with dedicated subsets and clear, distinct classification criteria. This distinction is critical for targeted evaluation of these fundamentally different risk types from the perspective of the computer security. |
> | **Scale & Comprehensiveness** | Limited data volume (569 records) and risk type coverage (10 types). | **Over 4x larger** (2,293 records) with significantly broader coverage (15 risk types, 29 scenarios), enabling more robust and less biased evaluation. |
> | **Evaluation Headroom & Challenge** | Nearing a performance ceiling; SOTA methods achieve near-perfect scores (~96% F1), limiting its ability to measure future progress. | Presents a **more challenging and discerning benchmark** with significant **headroom for improvement**, making it a more valuable tool for tracking future research progress. |
>
> As this table illustrates, ASSEBench is specifically designed to overcome the core limitations of prior work. It introduces entirely new evaluation dimensions—such as the nuanced handling of ambiguity and the clear separation of safety and security—that were previously unaddressed. By providing a more challenging benchmark with greater headroom, ASSEBench is not just a larger dataset, but a qualitatively superior tool designed to drive and measure meaningful progress in the field.
>
> ---
>
> ### **Q2: Positioning of AgentAuditor**
>
> Thank you for this insightful question about our work's positioning. We will expand the manuscript to better articulate our vision for AgentAuditor as a new paradigm for evaluation.
>
> Our motivation stems from a key observation: while LLMs show promise as evaluators, they have fundamental limitations. Standard methods, like prompt engineering or fine-tuning, treat the LLM as a monolithic reasoner. This makes them struggle with consistency, deep contextual understanding, and nuanced, ambiguous cases.
>
> Our work proposes a conceptual shift. Instead of trying to force all knowledge and rules into the LLM, we designed AgentAuditor as a **cognitive framework** around the LLM. It provides the LLM with our structured memory and a multi-stage retrieval mechanism. This method directly addresses the core weaknesses of LLM evaluators by externalizing the reasoning process, making it more explicit, verifiable, and consistent.
>
> Therefore, we do not position AgentAuditor as just another variant of an "LLM-based evaluator." We see it as a new paradigm for building **training-free, high-precision risk detection systems.** Its key advantages are:
>
> - **Training-Free and Adaptable:** It can be updated and improved simply by adding new experiences to its memory, avoiding the costly and data-intensive process of fine-tuning.
> - **High-Precision and Interpretable:** Its accuracy stems from being guided by highly relevant, structured precedents, and the retrieved CoT examples make its final judgment process more transparent.
>
> We will expand our introduction and conclusion to better articulate this. In the introduction, we will more clearly state our goal is to move beyond simple prompting towards a new paradigm of "memory-augmented reasoning" for evaluators. In the conclusion, we will add more discussion about the broader implications, suggesting that this framework of equipping LLMs with external, structured "experiential memory" could inspire future research into other complex, nuanced judgment tasks.
>
> ---
>
> ### **Limitations**
>
> Thank you for this suggestion. We agree that the main text should be more self-contained. We will restructure our final version by adding `Limitations` to the main text before `Conclusion & Future Work`. This will ensure that our work's boundaries are discussed transparently within the main body of the paper, improving its overall completeness.

---

### Official Review · Reviewer_Wyt9 · 2025-07-03

**Clarity:** 3
**Significance:** 4
**Originality:** 4
**Rating:** 5
**Confidence:** 3

**Summary:**

In the context of LLM-based agents, existing evaluators often overlook critical safety and security risks arising from agents’ autonomous actions. Different from generation scenarios, the agentic systems require deeper assessment of their safety and security. To address it, the paper introduces a reasoning framework, AgentAuditor, which leverages LLMs to extract structured semantic features and generate chain-of-thought (CoT) reasoning traces as experiential memory. These relevant traces are dynamically retrieved to assist LLM-based evaluators in assessing new cases. Furthermore, the authors propose a benchmark, ASSEBench, to evaluate how well LLM-based evaluators can identify and address safety and security risks in agentic behaviors.

**Questions:**

- How many total samples are used during reasoning memory construction, and how many representative shots are selected?
- How scalable is this process to larger datasets or more diverse domains?
- What if the retrieval component fails to find relevant reasoning memories for a given case? How robust is the framework in such scenarios?

**Ethical Concerns:**

["NO or VERY MINOR ethics concerns only"]

**Final Justification:**

The authors addressed my concerns in the rebuttal, and I encourage them to improve the final version accordingly. I have decided to keep my score at 5 (accept) as it is already positive.

**Limitations:**

N/A.

**Paper Formatting Concerns:**

N/A.

**Quality:**

3

**Strengths And Weaknesses:**

Strengths:
- The authors identify limitations in existing rule-based and LLM-based evaluators for agent safety and security, and propose a novel evaluation paradigm, AgentAuditor, to address these challenges.
- The authors also introduce a comprehensive benchmark, ASSEBench, which provides a valuable resource for systematically assessing the safety and security of LLM-based agents.
- The process of representatives sampling for reasoning memory construction is very interesting, which involves transformation of embeddings and hierarchy clustering.
- The proposed framework consistently improves evaluation performance across all datasets and models, and demonstrates its ability to emulate human expert evaluators.

Weaknesses:
- The reasoning memory is fixed after construction and depends on the chosen clustering method. Additionally, the paper lacks an analysis of how different clustering strategies might impact performance.
- The selection of heuristic weights used in the Reasoning Memory Construction process is not explained.
- The efficiency of the framework is not discussed. Components such as PCA and FINCH clustering may be computationally expensive, particularly when scaling to a larger sample set.
- Minors: The font size in figure text is too small for comfortable reading. There are some typos, such as at the end of line 131, and a missing “Stage 3” heading in Section 3.3.

---

> ### Author Rebuttal · Authors · 2025-07-29
>
> We are incredibly grateful for your insightful review. Your positive assessment is a great encouragement to us. Your suggestions will undoubtedly help us improve the quality and impact of our paper. We address your concerns below. Please note that Gemini-2-Flash-Thinking is used as the base model for all experiments presented in this rebuttal.
>
> ---
>
> ### **W1: Fixed Reasoning Memory and Clustering**
>
> Thank you for raising these two insightful points.
>
> For the first point, our reasoning memory is static after its initial construction. This is due to our primary research objective: enhancing the accuracy of the evaluator on existing benchmarks. For this task, building a high-quality, fixed memory through a one-time process is an efficient method that ensures consistency and reduces the computational cost.
>
> Your idea provide insightful inspiration for our future work. We plan to set a dynamic memory mechanism as one of the core directions for our future research and will discuss it in the revised version of this paper.
>
> Regarding the choice of the clustering method, we acknowledge that our method is dependent on the clustering method and our manuscript lacks a comparative analysis. In fact, during the development phase, we tested several clustering methods and ultimately selected FINCH for its superior overall performance.
>
> To address your concern, we have conducted new experiments on R-Judge, comparing FINCH with three classic clustering algorithms: K-Means, K-Medoids, and DBSCAN. For K-Means and K-Medoids, we set the number of clusters to 24 to match the output of FINCH for a controlled comparison. For DBSCAN, we adopt a standard heuristic approach to set its hyperparameters. The min_samples is set to 2*D (data dimensionality), and the eps is determined by identifying the elbow point of the k-distance graph. This procedure ultimately yields 14 shots.
>
> | Clustering | RM Shots | F1 | Acc |
> | :--- | :---: | :---: | :---: |
> | Base Model | 0 | 82.27 | 81.21 |
> | K-Means | 24 | 88.11 | 86.96 |
> | K-Medoids | 24 | 89.66 | 86.79 |
> | DBSCAN | 14 | 85.94 | 84.12 |
> | FINCH | 24 | 96.31 | 96.10 |
>
> Our experiments demonstrate that FINCH not only has the advantage of not requiring key hyperparameters but also achieves the best performance, thus justifying our selection. We will incorporate this experiment into the appendix to fully address your concerns.
>
> ---
>
> ### **W2: Unexplained Weights**
>
> Thank you for pointing out the omission of the weight selection process. This is a crucial point about clarity and transparency.
>
> To directly address your concern, we provide a brief explanation here. These weights are determined through a systematic and empirical search on an independent development set (R-Judge), on which we perform grid search. We predefine multiple sets of weight combinations and test each set on AgentAuditor and R-Judge. To minimize the effects of randomness, each test is repeated three times, and the results are averaged.
>
> We select the weight set that achieves the most optimal performance on R-Judge and adopt it as fixed parameters for our framework. To validate the generalization ability, we use these weights for all experiments in our paper.
>
> As demonstrated by our experiments (**Tables 1, 12, & 13**), our framework achieve significant performance improvements across all benchmarks. This indicates that our weights have good generalization capabilities.
>
> In the final version, we will add an appendix to detail this selection process increase the transparency and reproducibility of our work. Thank you again for your feedback!
>
> ---
>
> ### **W3: Lack of Efficiency Analysis**
>
> Thank you for the feedback. We acknowledge that a discussion on this part is missing from our manuscript.
>
> In our framework, the computational processes that involve only CPU, such as PCA and FINCH, do not constitute a performance bottleneck. Their cost is negligible compared to the core cost of LLMs.
>
> To address your concern, we conduct new experiments. We construct multiple simulated datasets, with scales ranging from 1x to 50x the original size, by duplicating R-Judge (564 records) and using LLMs to automatically modify its content. We test them on a server with two 64-core EPYC CPUs, measuring the used time. The results are shown below.
>
> | Scale | PCA Time (s) | FINCH Time (s) |
> | :--- | :---: | :---: |
> | 1x | 0.03 | 0.02 |
> | 5x | 0.14 | 0.09 |
> | 50x | 2.61 | 1.94 |
>
> The results demonstrate that the time cost is minimal and negligible compared to the total cost of our framework, as detailed in **Appendix P**. We will incorporate a detailed analysis into Appendix P.
>
> ---
>
> ### **Q1: The Number of Shots**
>
> Thank you for pointing out this issue. We acknowledge that, due to page limitations, this important information is not clearly presented in the main text, but is included in **Appendix L**. This is a structural issue. In the final version, we will move these details to the main text for better clarity.
>
> To directly answer your concern, the table below shows the total number of shots for each dataset and the number of shots selected for the reasoning memory. For representative shots, all experiments use fixed numbers: n=8 for candidate shots and k=3 for the final shots in reasoning.
>
> | Benchmark | Overall Shots | Shots in Reasoning Memory |
> | :--- | :---: | :---: |
> | ASSE-Safety | 1476 | 72 |
> | ASSE-Security | 817 | 73 |
> | ASSE-Strict | 1476 | 72 |
> | ASSE-Lenient | 1476 | 72 |
> | R-Judge | 564 | 24 |
> | AgentHarm | 176 | 19 |
> | AgentSecurityBench | 1600 | 71 |
> | AgentSafetyBench | 2000 | 78 |
>
> ---
>
> ### **Q2: Scalability**
>
> Thank you for this question. This is a key dimension for evaluating the utility of our framework. To address your concern, we analyze the scalability of AgentAuditor from two perspectives: dataset size and domain diversity.
>
> When scaling to larger datasets, our framework still requires only a one-time "memory construction" phase. Once the reasoning memory (RM) is built, the cost of evaluating each new case is constant and minimal, as it only depends on the small-scale RM, not the size of the original dataset. Furthermore, as shown in our response to W3, the computational overhead for AgentAuditor during the RM construction increases negligibly even as the dataset size grows.
>
> When scaling to diverse domains, our framework also has excellent extensibility due to its domain-agnostic design. The framework's adaptability stems from LLM-based zero-shot feature extraction and CoT generation, the unsupervised FINCH clustering, and a highly transferable RAG and memory system, rather than relying on any manually engineered, domain-specific rules. To apply our framework to a new domain, one only needs to provide a batch of interaction records, from which the framework can automatically construct a high-quality reasoning memory. This is supported by our experiments, where the framework achieves consistent and significant performance improvements across multiple benchmarks covering different scenarios. This validates its strong adaptation capabilities, allowing it to be migrated and extended to new, diverse domains at a low cost.
>
> In the final version of the paper, we will include a clear summary of this discussion on scalability in the main text to address your concerns.
>
> ---
>
> ### **Q3: Robustness**
>
> Thank you for this insightful question. We will address your question from both a mechanistic and an experimental perspective.
>
> Mechanistically, our retrieval component will not "fail" in the absence of similar shots, as our retrieval process is not based on a fixed similarity threshold but on ranking. For any given case, the system computes its similarity against all shots in the reasoning memory (RM) and always returns the top-k shots with the highest composite scores (k=3 typically). Therefore, even when a new case is out-of-distribution and all similarity scores are low, the system still provides k "least dissimilar" shots and corresponding CoTs to assist the evaluator.
>
> This shifts the core of the issue to your second question: how robust is the framework under these conditions?
>
> To answer this, we conduct a challenging cross-dataset experiment. We use the RM built from R-Judge to evaluate two entirely different datasets: ASSE-Safety and ASSE-Security. This effectively simulates the scenario where highly relevant shots are unavailable.
>
> | RM | Dataset | F1 | Acc |
> | :--- | :--- | :---: | :---: |
> | Base Model | ASSE-Safety | 61.79 | 67.82 |
> | R-Judge | ASSE-Safety | 86.36 | 86.10 |
> | ASSE-Safety | ASSE-Safety | 91.59 | 90.85 |
> | Base Model | ASSE-Security | 67.25 | 72.34 |
> | R-Judge | ASSE-Security | 84.55 | 85.12 |
> | ASSE-Security | ASSE-Security | 93.17 | 93.15 |
>
> While performance shows an expected decline when highly relevant shots are lacking, AgentAuditor still significantly outperforms the base model, demonstrating graceful degradation.
>
> To further validate the robustness of the 'memory-augmented reasoning' stage itself, we conduct a more rigorous ablation study on shot relevance. We compel the system to use less relevant shots (ranked 4th-6th) and compare this to using random shots and the baseline. The experiment is conducted on R-Judge.
>
> | Selection Method | F1 | Acc |
> | :--- | :---: | :---: |
> | AgentAuditor (Top 1-3 Shots) | 96.31 | 96.10 |
> | Ranks 4-6 Shots | 92.74 | 91.30 |
> | Random Shots | 85.07 | 82.80 |
> | Base Model | 82.27 | 81.21 |
>
> The experiments above provide strong evidence for the robustness of our framework. This suggests that even when the retrieved shots are not a perfect match, our framework still provides the LLM evaluator with a valuable logical structure, enabling it to perform significantly better than when "thinking alone."
>
> We will add these new experiments and their detailed analysis to the appendix of our paper to fully address your concerns and further demonstrate the robustness of our framework. Thank you again for your insightful feedback!

---

> > ### Comment · Reviewer_Wyt9 · 2025-08-05
> >
> > Thank authors for the rebuttal. It has addressed my concerns. I will maintain my previous scores.

---

### Comment · Area_Chair_cwLf · 2025-08-04

Dear Authors and Reviewers,

I would like to thank the authors for providing detailed rebuttal messages

To reviewers: I would like to encourage you to carefully read all other reviews and the author responses and engage in an open exchange with the authors. Please post your first response as soon as possible within the discussion time window. Ideally, all reviewers will respond to the authors, so that the authors know their rebuttal has been read.

Best regards,
AC

---

### Note · Authors · 2025-08-13

Dear Senior Area Chairs, Area Chairs, and Reviewers,

**Summary of Strengths Identified by Reviewers**
We sincerely thank the reviewers for their thoughtful review and are encouraged by the clear consensus on our key contributions:

*   **Clear Novelty and Significance:** Our work was thought to be a "novel evaluation paradigm" with **"excellent"** novelty and significance (Reviewer Wyt9), and praised as an **"important and timely research topic"** (Reviewer Fy7F).
*   **Substantial Benchmark Contribution:** Our benchmark was praised as a **"valuable resource for systematically evaluating the safety and security of LLM-based agents"** (Reviewer Wyt9).
*   **Strong Empirical Validation:** As noted by Reviewer Wyt9, our method **"consistently improve evaluation performance across all datasets and models"**, with Reviewer RJU9 confirming it outperforms baselines "by a large margin."

**Addressing Critical Feedback**

In response to constructive critique, we conducted new experiments to confirm our framework's **robustness** (via adaptive attacks), **adaptability** (via cross-dataset testing), and **racticality** (via time-cost analysis on larger datasets). We are pleased that these attempts successfully addressed the reviewers' concerns, as they attested in their comments.

**Our Pledge to Bettering**

We will perfect the final manuscript based on all feedback, particularly in:

*   **Improve the autonomy of the main paper.** We will integrate the most crucial findings in the appendix, e.g., time-cost analysis, and a distinct **Limitations**, into the main paper.
*   **Increase the positioning of contributions.** We will rephrase the introduction to better position AgentAuditor as a new evaluation paradigm and more clearly state the challenges in agent evaluation.
*   **Additional Experiments.** We will full information of new experiments to the appendix, including those on adversarial robustness, adaptability, and domain-specific performance.

**Final Remarks**

We appreciate the rigorous review which has significantly improved our work. We believe our detailed rebuttal and new experiments have fully addressed the initial concerns. We are encouraged by the fact that all three expert reviewers who particpated the rebuttal agreed with our paper, affirming our work's novelty and contributions. We believe our paper, buttressed by the majority of the experts and strengthened through rebuttal efforts, is a good contribution to the community.

---

### Decision · Program_Chairs · 2025-09-17

**Decision:**

Accept (poster)

**Comment:**

This paper presents AgentAuditor, a memory-augmented framework, and ASSEBench, a comprehensive benchmark, to address the critical challenge of evaluating LLM agent safety. The work was well-received by a majority of reviewers for its novelty, the significance of the problem, strong empirical results, and the value of its benchmark contribution.

Initial reviews raised several important questions regarding the framework's robustness to adaptive attacks, its adaptability, and efficiency. In response, the authors provided an thorough rebuttal, including a suite of new experiments that successfully addressed these concerns. This effort convinced the three engaged reviewers, who all confirmed their positive or borderline-positive ratings and acknowledged that their questions had been answered.

While concerns from one reviewer remained, the majority of the review committee was convinced by the authors' thorough rebuttal and new experiments, forming a consensus for acceptance.